# Prognostic Prediction for Therapeutic Effects of Mutian on 324 Client-Owned Cats with Feline Infectious Peritonitis Based on Clinical Laboratory Indicators and Physical Signs

**DOI:** 10.3390/vetsci10020136

**Published:** 2023-02-09

**Authors:** Masato Katayama, Yukina Uemura

**Affiliations:** Bloom Animal Hospital, Kajiyama 1-10-32, Tsurumi, Yokohama City 230-0072, Japan

**Keywords:** FIP, cat, nucleotide analogue, total bilirubin, neurologic clinical sign

## Abstract

**Simple Summary:**

Feline infectious peritonitis is a fatal disease classified into the effusive form, the non-effusive form, or their mixture showing clinical signs of both forms. To determine whether the therapeutic effect of Mutian on the non-effusive and the mixed forms of feline infectious peritonitis can be predicted using clinical indicators before starting treatment, we entered 161 cats with the mixed form and 163 cats with the non-effusive form into this study. Physical assessment, detection of viral genes, and several clinical laboratory tests were performed before Mutian was administered. These indicators were compared between the disease groups that survived after receiving Mutian for 84 days and those that died before the completion of treatment. Significant differences in body temperature, appetite scores, and activity scores were confirmed between the surviving and non-surviving groups. The therapeutic effect was insufficient when total bilirubin levels increased in cats with the mixed form. In both forms of feline infectious peritonitis, therapeutic effects were difficult to obtain when neurological clinical signs had been observed. The therapeutic effects of Mutian on the cats with these forms of feline infectious peritonitis can be predicted based on pre-treatment body temperature, appetite scores, and activity scores, as well as the presence of neurological signs.

**Abstract:**

Feline infectious peritonitis (FIP) is a fatal disease classified as either effusive, non-effusive (‘dry’), or a mixture (‘mixed’) of the forms of FIP, with mixed showing signs of both effusive and dry. To determine whether the therapeutic effect of Mutian on dry and mixed FIP can be predicted using clinical indicators before starting treatment, we entered 161 cats with mixed FIP and 163 cats with dry FIP into this study. Physical assessments, the reverse transcriptase-PCR detection of viral genes, and clinical laboratory tests (hematocrit, albumin/globulin ratio, serum amyloid A, α1-acid glycoprotein, and total bilirubin) were performed before Mutian was administered. These indicators were compared between the FIP groups that survived after receiving Mutian for 84 days and those that died before the completion of treatment. Significant differences in body temperature, appetite, and activity scores were confirmed between the surviving and non-surviving groups. The therapeutic effect was insufficient when total bilirubin levels increased in cats with the mixed form. In both of the FIP types, therapeutic effects were difficult to obtain when neurological clinical signs were observed. The therapeutic effects of Mutian on the cats with dry and mixed FIP can be predicted based on pre-treatment body temperature, appetite scores, and activity scores, as well as the presence of neurological signs.

## 1. Introduction

Feline infectious peritonitis (FIP) is a fatal feline disease caused by an excessive pro-inflammatory response to feline coronavirus (FCoV) [1,2,3]. In the early disease stages, FCoV-infected cats exhibit non-specific clinical signs such as recurrent fever, vomiting, and diarrhea [1,3]. FIP is classified as effusive (wet), non-effusive (dry), or a mixture (mixed) of both forms [1,2]. The wet form is characterized by fibrinous pleural peritonitis with vasculitis, leading to fluid redistribution into neighboring spaces, and effusion accumulation in body cavities including abdominal, thoracic, pericardial and scrotal; the dry form is characterized by granulomatous lesions in several organs, including the central nervous system, and ocular clinical signs; characteristics of both types are observed simultaneously in the mixed form [1,4]. Currently, ante-mortem diagnosis of FIP is difficult; furthermore, there are no definitive, non-invasive diagnostic tests for cats without effusion. In routine veterinary practice, the disease is diagnosed based on the results of physical examination, clinical laboratory tests, and viral RNA detection [4,5].

Currently, there is no authority-approved therapy for FIP and only supportive care is recommended which aims at prolonging survival [6]. Although several studies have reported the use of immune stimulants or the other related agents, thus far their clinical advantages have not been verified [7,8]. Therefore, avoiding group hoarding and reducing stress in infected cats remain the predominant measures to relieve symptoms [1,4,5,6].

In 2018, the nucleoside analog GS-441524 was demonstrated to suppress feline infectious peritonitis virus (FIPV) [9]. Its prodrug was found to inhibit the replication of several taxonomically diverse RNA viruses [10]. The efficacy of GS-441524 against non-neurological FIP was proven based on an >80% recovery from disease onset [11]. Moreover, GS-441524 administration at higher doses was found to be effective in cats with ocular and neurological FIP [12].

Based on the potential effects of nucleoside analogs, such as GS-441524, on FIP, an oral preparation, Mutian^®^ Xraphconn (Mutian X), was developed for veterinary healthcare use (Mutian Life Sciences Co., Ltd., Nantong, China) [13]. The main drug component of Mutian X, GS-441524, exerts a remarkable therapeutic effect on FIP, including the wet type [14,15,16]. We previously administered Mutian X to 141 cats with wet-type FIP, observed its promising therapeutic effect, and confirmed remission in approximately 82% of the cats [17]. Furthermore, we showed that its effect can be predicted using circulating bilirubin levels as a prognostic indicator prior to treatment initiation [17]. In the present study, we determined the therapeutic effect of Mutian administration on dry and mixed types of FIP, and predicted its effect by analyzing various clinical diagnostic parameters before administration.

## 2. Materials and Methods

### 2.1. Drugs and Therapeutic Protocol

Mutian X was obtained from Mutian Life Sciences Co., Ltd. (Nantong, China). Based on previous studies, the main active ingredient of Mutian X is GS-441524, and the Mutian X 100 mg formulation includes 5 mg of GS-441524 [16,17]. The active ingredient of the injectable product, designated as Mutian II, has not yet been confirmed by third parties, but is presumably identical to that of Mutian X for oral administration.

Between June 2019 and December 2021, 161 cats diagnosed with mixed-type FIP and 163 cats diagnosed with dry-type FIP at our hospital were included in the study. Mutian X was administered to the cats as described previously [17]. To put it briefly, immediately after the initial FIP diagnosis, Mutian was administered at q24h for mixed-type FIP at 150 mg/kg in the early to middle disease stages or at 200 mg/kg in the late stage. The drug was administered orally or by subcutaneous injection at 130 mg/kg at q24h for dry-type in the early stage, 150 mg/kg in the middle stage, or 200 mg/kg in the late stage. All drug administrations were continued from day 0 to 84. In the dry-types, the cats were classified into the “early” phase if they were under 2 years of age, with persistent fever, depression, anorexia, weight loss, stunted growth, deterioration of coat, no response to antibiotics, diarrhea or constipation, disease-onset-driven leukocytosis and neutrophilia, lymphopenia, increased serum total protein, hyperglobulinemia and hypoalbuminemia [albumin-to-globulin ratio (A/G) ≤ 0.6], formation of granulomas on organs, swelling of the liver, kidneys, testicles, and mesenteric lymph nodes, peritoneal inflammation, and ring formation in the renal medulla. The cats were classified into the “middle” phase if they showed some progression of the early symptoms, chronic non-regenerative anemia [hematocrit (HCT) ≤ 24%], a change in urine color to a dark yellow due to hyperbilirubinemia, and ocular lesions including uveitis (anterior aqueous humor, clouding of the iris, obscuring of the iris pattern, formation of microscopic blood clots, and appearance of intravascular granulomatous nodules). The cats were classified into the “late” phase if they showed a rapid deterioration of physical condition, severe anemia (HCT ≤ 16%), anorexia, and neurological symptoms (including nystagmus, resting muscle tremors, hind-limb weakness, decreased jumping ability, body stiffness, slow movements, and postural instability). The mixed-types manifest both the wet and dry forms of FIP simultaneously, and their staging follows the dry-type criteria. These classifications were determined by a veterinarian based on the above criteria, and considering an interview with the owner during the first examination, along with the results of various clinical laboratory tests. Tablets or capsules were used for oral administration. If oral administration was difficult owing to gastrointestinal dysfunction or FIP onset, such as an inability to absorb nutrients, subcutaneous administration was performed at the same dose. The drug was administered on a regular basis and on an empty stomach [17]. Each patient underwent a standard drug administration protocol for 84 days, followed by a 3-month observation period; stable cases with no apparent deterioration in quality of life were regarded as being in remission [17]. Oral-administration was performed by the cat’s owner. Temporary administration of the injectable form (Mutian II) required hospitalization at our facility, but long-term administration was performed under the supervision of a veterinarian near the owner’s home.

### 2.2. Patients and Diagnostic Procedures

The cases of FIP were diagnosed and classified at initial examination into 161 and 163 cases of mixed and dry FIP, respectively. The transition stages from wet to dry, or dry to wet, were not monitored in this study. Thirty cats suspected of having FIP from June 2019 to December 2021 were eventually diagnosed with a disease other than FIP (non-FIP subjects). Their detailed clinical signs are described in the Results section. All these cases (354 cases in total, including 161 cases of mixed FIP and 163 cases of dry FIP) were included in our study. Informed consent was obtained from the cats’ owners by assuring in advance that Mutian would be administered to all FIP cats under optimal conditions for treatment as standard of care, and not as an experimental therapy. Ethical review and approval were waived because the above informed consent acquisition was legally implemented in accordance with the declaration of the Japanese Veterinary Medical Association and all data were obtained within the scope of usual veterinary care and anonymized appropriately. The owners of all cats agreed to the use of their biological samples and test results in the study.

Clinical specimens (whole blood with EDTA or plasma samples isolated by centrifugation of heparinized whole blood) were collected from all cats at the time of initial medication and after the completion of Mutian treatment (84 days after the first medication). The body cavity effusions were collected from most cats with mixed FIP. Measurements of total bilirubin (TB), A/G, serum amyloid-A (SAA), HCT, and α1-acid glycoprotein (α1AG) concentrations in plasma, and reverse transcriptase-PCR (RT-PCR) tests were performed as described in a previous study [17]. Furthermore, SAA levels exceeding the upper limit of the measurable range were considered as 225 μg/mL for statistical analysis. Cats with circulating α1AG levels exceeding the upper limit of the normal range (736 μg/mL) were considered positive [17].

The samples were considered positive for FCoV when the target gene was detected in blood or body cavity effusion by RT-PCR technology according to previous methods [18]. The parameters of appetite (volume, frequency, and speed of feed intake) and activity (movement, walking speed, and agility) were assessed at the time of the initial interviews with the owners. An appetite score was determined by interviewing the cat’s owner according to predetermined criteria. An activity score was also determined from five interview items, and resultantly classified into six stages, from 0 to 5. Each parameter was then converted to an arbitrary value as an appetite or activity score on a scale of 1–6; the scores were then analyzed statistically to assess the owner’s perception of the cat’s physical condition, along with the results of the routine physical examination at the clinic (body temperature, weight, echography, auscultation, and palpation), according to the method described in our previous study [17]. Ultrasonography was performed in order to confirm and detect the accumulation of ascites or pleural effusions or swollen lymph nodes in the cats using an Aplio a CUS-AA000V ultrasound system (Canon Medtech Supply Co., Kanagawa, Japan).

The disease characteristics were confirmed through a comprehensive examination of the apparent clinical signs (anorexia, underactivity, vomiting, diarrhea, seizures, tremors, ataxia, or others), qualitative PCR-based detection of FCoV in the blood, ascites, and pleural effusions, and laboratory tests (HCT, whole cell count, total protein, TB, A/G, SAA, and α1AG). The age, body temperature, and weight of all cats were recorded during the initial consultation with the veterinarian. The diagnosis and sample collection of ascites, pleural effusion and abdominal lymphadenopathy in the internal organs of each case were performed by endoscopic ultrasound-guided fine-needle aspiration [19,20]. Neurological signs (e.g., ataxia, head tilt, hyperesthesia, nystagmus, seizures, and behavioral changes) and ocular signs (e.g., iritis, corneal edema, dyscoria/anisocoria, loss of vision, hyphema, hypopyon, keratic precipitates, aqueous flare, perivascular cuffing, chorioretinitis, and subretinal fluid accumulation causing partial retinal detachment) characteristic of FIP were monitored [21], and the veterinarians determined the presence or absence of each clinical sign based on the neurological examination sheet of the Japanese Society of Veterinary Neurology [22]. As reported previously, obtaining a definitive diagnosis of FIP based on non-invasive approaches is difficult [21]. Under such a circumstance, in order to make a diagnosis of FIP in our routine healthcare, we made a diagnosis by a comprehensive analysis of the above indicators. Thirty cats with non-FIP diseases were judged not to have FIP during the diagnostic process described in the above section.

### 2.3. Statistical Analysis

Numerical indicators, including age, appetite score, activity score, body temperature, body weight, HCT, A/G, TB, and SAA, were compared between the survivor and non-survivor groups using the Mann–Whitney non-parametric U test. The presence of diarrhea, vomiting, FCoV gene expression, and significant α1AG levels were classified as categorical data and analyzed using Fisher’s exact test as 2 × 2 contingency tables with any expected cell values below 5. We also set up a non-FIP group of 30 cats to clarify the clinical characteristics of mixed and dry FIP, and judged that the Mann–Whitney non-parametric U test between two unpaired groups is suitable for testing the significant difference between the two groups. In addition, we determined that Fisher’s exact test is appropriate for the analysis of the significance level in differences between two groups of categorical data. All the obtained numerical data were shown in the tables as mean values and standard errors (SEs). We also measured four numerical parameters, namely body weight, HCT, A/G, and SAA levels, in the surviving cats treated with Mutian. Differences in the values before and after drug administration were assessed using the Wilcoxon signed-rank test. The distribution of each parameter before and after standard treatment with Mutian medication was plotted as a box-and-whisker diagram using Excel 2016 (Microsoft). A *p* value of < 0.05 was considered significant. All statistical analyses were performed using StatView 5.0 (SAS Institute, Cary, NC, USA).

## 3. Results

### 3.1. Comparison of Parameters between FIP and Non-FIP Groups

The results of the clinical parameters in mixed FIP cats (*n* = 161) and dry FIP cats (*n* = 163) compared with those in non-FIP cats (*n* = 30) are summarized in Table 1 and Table 2. The age of the cats enrolled in this study ranged from 2 months to 13.6 years. The ratios of mixed and dry FIP disease below 2 years for each of the total subjects were 85.1% and 81.6%, respectively. The proportion of crossbreeds among the mixed and dry FIP cats was 24.8% and 41.7%, respectively; the purebreds mainly comprised Russian Blue, Norwegian, Bengal, Ragdoll, and American Shorthair cats. The non-FIP group (*n* = 30) showed upper respiratory infection (*n* = 6); diarrhea and lymphoma (*n* = 3 each); gastroenteritis, epithelial tumor, and chronic kidney disease (*n* = 2 each); and stomatitis, renal failure, encephalitis, thrombocytopenia, lymphadenopathy, traumatic bleeding, lameness, side effects of vaccination, hind-limb femoral artery thromboembolism, feline leukemia virus infection, feline immunodeficiency virus infection, hepatobiliary pancreatitis, intestinal coronavirus infection, and elevated liver enzyme levels (*n* = 1 each).

There were no significant differences in age, body temperature, or weight between cats with and without mixed FIP or between cats with and without dry FIP. However, the appetite score (*p* < 0.0005), activity score (*p* < 0.0003), HCT (*p* < 0.0001), and A/G (*p* < 0.0001) were significantly lower in cats with mixed FIP than in those without FIP (Table 1). Moreover, the SAA levels in cats with mixed FIP were significantly higher (*p* < 0.002) than those in cats without FIP (Table 1). In contrast, appetite score, activity score, HCT level, and A/G in the dry FIP cats seemed to be lower than those observed in the non-FIP cats; however, only one parameter, A/G, was confirmed to have a statistically significant difference (Table 1, *p* < 0.0001). The levels of SAA in the dry FIP group tended to be higher than those in the non-FIP group, but the difference was not statistically significant (Table 1). The ratios of high α1AG were significantly elevated in both mixed (*p* < 0.0001) and dry FIP (*p* < 0.006), compared with those in the non-FIP group (Table 2). The TB levels seemed to be higher in the mixed and lower in the dry FIP groups than in the non-FIP group, but no statistically significant difference was found (Table 1).

There was no significant difference in the incidence of diarrhea, vomiting, neurological signs, ocular lesions, and abdominal lymphadenopathy between the mixed FIP and non-FIP cases (Table 2). However, in the comparison between dry FIP and non-FIP groups, a slightly significant difference was detected between the two groups in the incidence of diarrhea and abdominal lymphadenopathy (Table 2).

The positive rate of FCoV gene detection in blood using RT-PCR was higher in both the mixed and dry FIP groups than in the non-FIP group (Table 2, both *p* < 0.0001). Significantly higher rates of ascites and pleural effusion retention were detected in the mixed FIP group than those in the non-FIP group (Table 2; *p* < 0.0001 and *p* < 0.02, respectively). Furthermore, of the 113 cases for which body cavity effusions could be collected and RT-PCR testing was possible, 108 were positive for viral RNA, indicating that the positive rate of FCoV gene detection by PCR in either ascites or pleural effusion was 95.6% (Table 2, 108 positives/113 total).

### 3.2. Comparison of Parameters between Surviving and Non-Surviving Cats with FIP

In the present study, Mutian was administered to 161 pet cats with mixed FIP and 163 pet cats with dry FIP according to the standard treatment schedule for 84 days. Of these, approximately 85.1% (137/161) and 93.9% (153/163) of the cats survived, respectively (Table 3, Table 4, Table 5 and Table 6), with a substantially improved quality of life (Figure 1 and Figure 2). In 262 of 324 cases with mixed and dry FIP, Mutian X was administered orally, and only one patient received subcutaneous injections of Mutian II alone over the treatment course. Subcutaneous administration was feasible only for a short period in the early stages of FIP, and was preferred only in cases where oral administration was difficult because of disease progression. However, treatment could not be continued due to clinical deterioration, and almost all such cats died. As reported in our previous study, the therapeutic effect of the subcutaneous administration of Mutian II could not be verified because of the small number of subjects [17].

**Table 3 vetsci-10-00136-t003:** Statistical comparison of signalment and clinical parameters between surviving and non-surviving cats with mixed FIP.

Parameters	Surviving			Non-Surviving			*p*-Value
	*n*	Mean	SE	*n*	Mean	SE	
Age (months)	137	15.58	2.21	24	24.54	6.35	<0.02
Body weight (kg)	137	2.60	0.09	24	2.72	0.19	NS
Body temperature (℃)	123	38.88	0.09	19	37.47	0.60	<0.006
Appetite score	137	3.18	0.11	24	1.96	0.22	<0.0001
Activity score	137	3.19	0.10	24	2.13	0.21	<0.0001
HCT (%)	137	23.46	0.57	24	21.80	1.67	NS
A/G	136	0.44	0.01	23	0.43	0.02	NS
SAA (μg/mL)	136	113.85	6.23	22	96.13	14.73	NS
TB (mg/dL)	103	1.31	0.15	19	3.42	0.72	<0.0004

Differences are judged to be statistically significant at *p* < 0.05, using Mann–Whitney non-parametric U-test. Abbreviations: FIP, feline infectious peritonitis; SE, standard error of the mean; NS, not significant; HCT, hematocrit; A/G, albumin-to-globulin ratio; SAA, serum amyloid-A; TB, total bilirubin.

**Table 4 vetsci-10-00136-t004:** Statistical comparison of categorical parameters between surviving and non-surviving cats with mixed FIP.

	Surviving			Non-Surviving			*p*-Value
	*n*	Positive	Negative	*n*	Positive	Negative	
Diarrhea	137	30	107	24	6	18	NS
Vomiting	137	12	125	24	4	20	NS
Neurological signs	137	31	106	24	15	9	<0.0004
Ocular lesions	137	10	127	24	1	23	NS
Abdominal lymphadenopathy	137	34	103	24	2	22	NS
α1AG *	135	134	1	19	19	0	NS
PCR testing (blood)	135	121	14	21	17	4	NS
PCR testing (ascites or pleural) **	97	92	5	16	16	0	NS

Differences are judged to be statistically significant at *p* < 0.05, using Fisher’s exact test. * The samples were considered positive if the α1AG level exceeded the upper limit of the normal range (736 μg/mL). ** The samples were considered positive for FCoV when the target gene was detected in both or either ascites or pleural effusions. Abbreviations: FIP, feline infectious peritonitis; NS, not significant; PCR, polymerase chain reaction; α1AG, α1-acid glycoprotein.

**Table 5 vetsci-10-00136-t005:** Statistical comparison of signalment and clinical parameters between surviving and non-surviving cats with dry FIP.

Parameters	Surviving			Non-Surviving			*p*-Value
	*n*	Mean	SE	*n*	Mean	SE	
Age (months)	153	17.79	1.77	10	11.60	3.90	<0.03
Body weight (kg)	153	3.00	0.08	10	1.80	0.23	<0.0003
Body temperature (℃)	131	38.91	0.07	6	36.97	0.56	<0.004
Appetite score	153	3.62	0.13	10	2.10	0.56	<0.007
Activity score	153	3.84	0.12	10	1.90	0.39	<0.0006
HCT (%)	153	29.82	0.69	10	26.52	2.53	NS
A/G	152	0.49	0.01	10	0.50	0.04	NS
SAA (μg/mL)	148	82.77	5.70	9	59.89	17.90	NS
TB (mg/dL)	91	0.73	0.11	8	1.70	0.52	NS

Differences are judged to be statistically significant at *p* < 0.05, using Mann–Whitney non-parametric U-test. Abbreviations: FIP, feline infectious peritonitis; SE, standard error of the mean; NS, not significant; HCT, hematocrit; A/G, albumin-to-globulin ratio; SAA, serum amyloid-A; TB, total bilirubin.

**Table 6 vetsci-10-00136-t006:** Statistical comparison of categorical parameters between surviving and non-surviving cats with dry FIP.

	Surviving			Non-Surviving			*p*-Value
	*n*	Positive	Negative	*n*	Positive	Negative	
Diarrhea	153	32	121	10	3	7	NS
Vomiting	153	12	141	10	3	7	NS
Neurological signs	153	37	116	10	6	4	<0.03
Ocular lesions	153	27	126	10	3	7	NS
Abdominal lymphadenopathy	153	59	94	10	4	6	NS
α1AG *	148	137	11	7	6	1	NS
PCR testing (blood)	145	117	28	8	7	1	NS

Differences are judged to be statistically significant at *p* < 0.05, using Fisher’s exact test. * The samples were considered positive if the α1AG level exceeded the upper limit of the normal range (736 μg/mL). Abbreviations: FIP, feline infectious peritonitis; NS, not significant; PCR, polymerase chain reaction; α1AG, α1-acid glycoprotein.

**Figure 1 vetsci-10-00136-f001:**
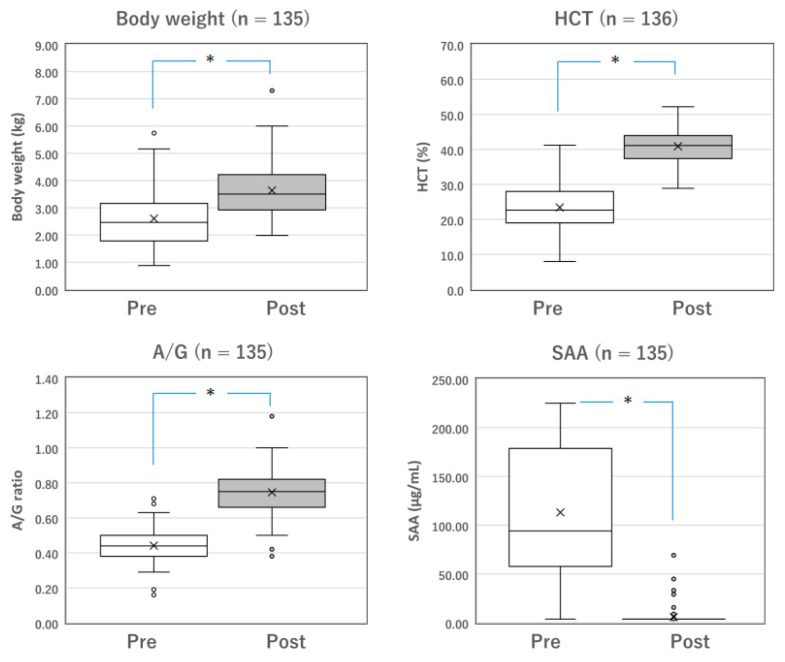
Changes in the parameters of surviving cats with mixed FIP before and after Mutian treatment. The body weight, HCT, and A/G of the surviving cats were significantly improved after Mutian therapy (Post) compared with their values at the initial examination prior to drug therapy (Pre) (* *p* < 0.0001). In contrast, SAA levels were drastically decreased after therapy (* *p* < 0.0001). The ‘×’ and the horizontal line in the box represent the mean and median, respectively. The median divides the box into interquartile ranges, and the box represents 50% of the data set, distributed between the 1st and 3rd quartiles. The whiskers show the outlier range outside the upper and lower quartiles. Open circles denote outliers. All statistical analyses were performed using the Wilcoxon signed-rank test (non-parametric). HCT, hematocrit; A/G, albumin-to-globulin ratio; SAA, serum amyloid-A.

**Figure 2 vetsci-10-00136-f002:**
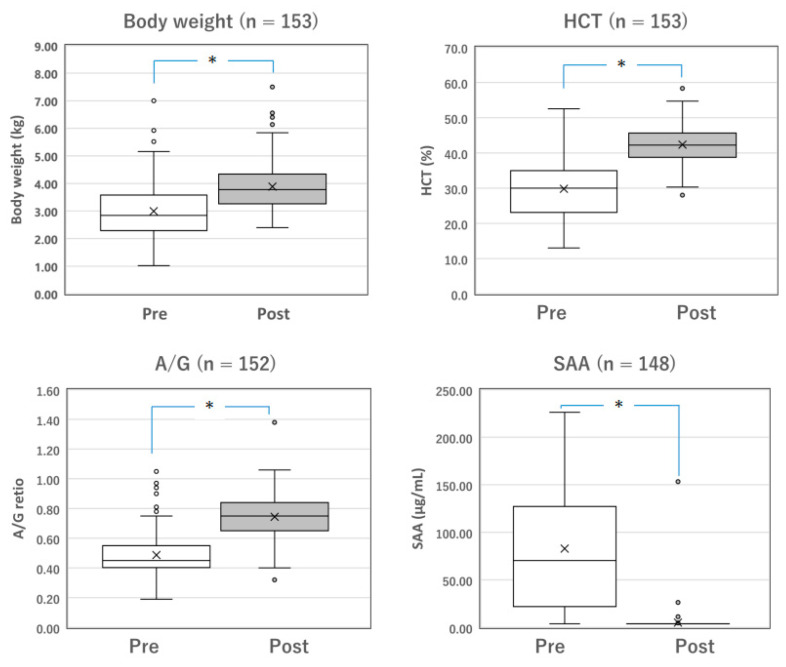
Changes in the parameters of surviving cats with dry FIP before and after Mutian treatment. The body weight, HCT, and A/G of the surviving cats were significantly improved after Mutian therapy (Post) compared with their values at the initial examination prior to drug therapy (Pre) (* *p* < 0.0001). In contrast, SAA levels were drastically decreased after therapy (* *p* < 0.0001). The ‘×’ and the horizontal line in the box represent the mean and median, respectively. The median divides the box into interquartile ranges, and the box represents 50% of the data set, distributed between the 1st and 3rd quartiles. The whiskers show the outlier range outside the upper and lower quartiles. Open circles denote outliers. All statistical analyses were performed using the Wilcoxon signed-rank test (non-parametric). HCT, hematocrit; A/G, albumin-to-globulin ratio; SAA, serum amyloid-A.

Furthermore, in the mixed FIP group, we investigated the differences in age, weight, temperature, appetite score, activity score, and clinical laboratory parameters (HCT, A/G, SAA, and TB levels) between the group that survived after receiving standard treatment with Mutian for 84 days (*n* = 137) and the group that died before the completion of drug therapy (*n* = 24). Although a slightly significant difference was detected in age (*p* < 0.02), no differences were observed in body weight, HCT, A/G, and SAA levels between the two groups (Table 3). However, the body temperature, appetite, and activity scores before therapy were significantly lower in the non-surviving group than in the surviving group (*p* < 0.006, *p* < 0.0001, and *p* < 0.0001, respectively). Furthermore, a significant increase (approximately 2.6 times on average) in TB levels was observed in the non-surviving group compared with those in the surviving group (*p* < 0.0004) (Table 3). Furthermore, no significant difference was observed in cases that were positive for diarrhea, vomiting, eye lesions, and abdominal lymphadenopathy between the surviving and non-surviving groups among the mixed FIP cases (Table 4). Comparative analysis between the surviving and non-surviving groups in the mixed FIP cats resulted in common endpoints, which were also observed in our previous study on wet FIP [17].

In contrast, in the dry FIP cases, no significant difference was detected in the levels of HCT, A/G, and SAA between the surviving (*n* = 153) and non-surviving groups (*n* = 10) in the dry FIP cases (Table 5). In addition, body temperature, appetite, and activity scores were significantly lower in the non-surviving group than in the surviving group in the dry FIP cases, similar to those in the mixed FIP cases (*p* < 0.004, *p* < 0.007, and *p* < 0.0006, respectively). An average increase of >2 times in TB levels was observed in the non-surviving group compared with that in the surviving group, but no statistically significant difference was found (Table 5). However, the non-surviving group in the dry FIP cases was younger in age and their body weight was lower than that in the surviving group, in contrast to that in the mixed FIP group (Table 5, *p* < 0.03 and *p* < 0.0003).

Interestingly, in both the mixed and dry FIP cases, neurological clinical signs were observed at a significantly higher frequency in the non-surviving group (15 positives/24 total and 6 positives/10 total, respectively) and were more prominent in the mixed group (*p* < 0.0004 in Table 4 and *p* < 0.03 in Table 6). No significant differences were detected in any of the other category endpoints between the surviving and non-surviving groups among the mixed and dry FIP cases (Table 4 and Table 6).

Within four weeks of the completion of standard drug administration, 11 cases relapsed showing multiple clinical signs, including anorexia, hypoactivity, fever, neurological signs, ascites, and pleural effusion. For these relapsed cases, we administered an additional course of Mutian X (200 mg/kg, q24 for 42 days).

### 3.3. Changes in the Parameters of Surviving Cats with FIP before and after Treatment

Among the surviving cats with mixed FIP (*n* = 137) and those with dry FIP (*n* = 153), changes in measurable parameters, including body weight, HCT, A/G, and SAA levels, were compared before and after the completion of standard Mutian therapy (Figure 1 and Figure 2). The body weight, HCT, and A/G of the surviving cats were significantly improved after drug treatment compared with their values at the initial examination (Figure 1 and Figure 2, both *p* < 0.0001). SAA levels drastically decreased after drug treatment (Figure 1 and Figure 2, both *p* < 0.0001).

## 4. Discussion

The adenosine nucleoside analog GS-441524 has been shown to be significantly effective against FCoV, and its therapeutic efficacy against naturally occurring FIP has been demonstrated in recent studies [7,8,9,10,11,12]. Mutian X was developed as an oral preparation and contains similar nucleic acid analogs for veterinary use; its apparent therapeutic effects have already been confirmed in cats with FIP [14,15]. According to the commercial drug package, the active ingredient of Mutian X is MT0901. The molecular formula, weight, and pharmacological activity of MT0901 are identical to those of GS-441524, and chemical and structural analyses have shown that the active ingredient in Mutian X is GS-441524 [16]. As the first step, efficient pharmacological treatment for FIP is now available, and therefore, we reasonably consider that diagnostic indicators for curable disease groups are important to maximize the therapeutic effects of drugs in the next process [3].

Our previous study investigated the therapeutic effects of Mutian X in 141 pet cats with wet FIP under diverse dietary and environmental conditions, and found that approximately 80% of cats with effusive FIP could be treated and achieve an improved quality of life after Mutian X administration, consequently revealing a statistically significant therapeutic effectiveness for the disease [17]. It is well known that wet FIP accounts for the majority of all FIP disorders [1,3]. There has been a previous report describing therapeutic efficacy with Mutian X in a cat with non-effusive FIP, but he had no neurological signs, only ocular and systemic signs [15]. Our present study is the first report to show a major recovery in neurological cases with Mutian treatment. We also observed survival rates of 85.1% (137 cured/161) in mixed FIP cats and 93.9% (153 cured/163) in dry FIP cats at 84 days after Mutian administration, both of which were regarded as superior to those in the wet FIP cases observed in our previous study.

Although the results of our comparative study of physical and clinical laboratory parameters in mixed FIP with those in the non-FIP disease were similar to our previous results with 141 wet FIP cats, only minor statistically significant differences were found in each of the parameters of dry FIP compared with those in the non-FIP cases (Table 1) [17]. Considering the high survival rate of dry FIP cats treated with Mutian, the dry FIP considered in this study may be less severe than the other two types of FIP. In the RT-PCR analysis using ascites or pleural effusion for the mixed FIP cases in this study, FCoV genes were detected in most cases (108/113) (Table 2). Immune complexes comprising anti-FCoV antibodies (produced as a host defensive reaction) and FCoV antigens may be deposited in blood vessels, causing inflammatory vasculitis in capillaries throughout the body, and leakage of fluid which accumulates as ascites or pleural effusions in the abdominal or thoracic cavity, respectively [19]. In mixed FIP, SAA levels, known as a clinical parameter for inner-vessel inflammation, which is thought to induce effusions, were elevated (*p* < 0.002) compared with those in the non-FIP cases, but these increases in the dry FIP cases were not significant, presumably suggesting that the inflammatory reaction including vasculitis, could not be enhanced and the accompanying appetite and activity reductions were also not significant in dry FIP (Table 1).

Levels of TB, A/G, SAA, and α1AG are correlated with the onset or severity of FIP [3,4,23,24,25,26,27]. These clinical laboratory parameters were also monitored for each case in our previous study on the treatment of 141 wet FIP cases, and the same analysis was performed in the present study with mixed and dry FIP cats. To validate the therapeutic effect of Mutian on mixed (*n* = 161) and dry (*n* = 163) FIP cats and to predict their prognosis, multiple parameters were measured at our initial examination, and statistical analysis of numerical or categorical items was performed between the group that survived because of Mutian treatment and the group that did not respond to treatment and could not survive (Table 3, Table 4, Table 5 and Table 6). Similar to our previous study, body temperature, appetite, and activity scores were significantly decreased in the non-surviving group compared with those in the surviving group after Mutian therapy, and were shown to provide subsidiary information on predicting its therapeutic effect on the mixed and dry FIP (Table 3 and Table 5). Similar to our previous results on wet FIP cases, we also observed that in the present study, both the pre-treatment TB levels in the mixed and the dry FIP tended to be higher in the non-surviving group than those in the surviving group. Although the difference in TB levels between the surviving and non-surviving groups among the mixed FIP cases was found to be remarkably significant (*p* < 0.0004), no significant difference was found in the dry FIP cases, suggesting that the prognosis for the success of Mutian treatment by TB levels may be exclusive to effusive FIP, including the wet and mixed disease.

A previous study reported frequently occurring hyperbilirubinemia in wet FIP [23]. Another study reported that extremely elevated TB levels were noted in cats with FIP from 2 weeks to 0 days before death in serial blood examinations [28]. Thus, the classification of disease status for effusive FIP (including wet and mixed types) based on the initial TB level as a predictor of survival may offer valuable information to veterinarians and cat owners while making decisions regarding cat treatment. Although parenchymal liver disease is known to cause hyperbilirubinemia in cats, increased erythrocyte fragility leading to hemolysis along with decreased clearance of hemoglobin-derived products may also be an underlying reason, especially in cats with FIP [4,23].

The physical characteristics of mixed and dry FIP cats include neurological clinical signs, ocular lesions, and lymph node enlargement [1,3,19,21]. We assessed significant differences in the frequency of these clinical signs between surviving and non-surviving groups among both mixed and dry FIP cats, and observed a tendency for lower survival in both FIP types when neurological signs were identified (Table 4 and Table 6, *p* < 0.0004 and *p* < 0.03). For the majority of cats showing neurological disease, symptoms are caused by FCoV-induced hydrocephalus [29,30]. The main active ingredient of Mutian X is a nucleic acid analogue with anti-viral effects. Its pharmacokinetic analysis in cats showed that its cerebrospinal fluid concentration was approximately 20% of its plasma levels and that doses five times those shown to effectively treat non-neurologic FIP would be necessary to achieve the drug concentration required to prevent viral cytopathic effects [31]. The results obtained in our study suggest that the possible spread of FCoV infection to the central nervous tissue, where anti-viral substances may be inadequately distributed, may result in increased disease severity and lower survival rates (Table 4 and Table 6).

In both mixed and dry FIP cats, body weight, HCT, and A/G were significantly improved following treatment with Mutian, as were the reductions in SAA levels, all of which indicate improved quality of life (Figure 1 and Figure 2). We observed that these results are similar to the results we previously observed for 116 cats with wet FIP, and also significantly confirmed that these four parameters can be utilized as monitoring indicators to estimate the therapeutic effectiveness of Mutian in all types of FIP [17].

The first limitation of this study is that the number of non-surviving cats with dry FIP was extremely small (*n* = 10). In future studies, it is necessary to include more feline subjects to obtain a statistically more reliable comparison between surviving and non-surviving groups among dry FIP cases.

The 30 cats with non-FIP diseases used as control subjects in this study were a completely different population from the 28 non-FIP cats enrolled in our previous study with wet FIP, to avoid repeated use of the same animals. As we did not use identical non-FIP subjects as controls, we could perform the same analysis as in our previous study on wet FIP cats. The second limitation of our study is the difficulty in collecting appropriate non-FIP cats. The current control group contained a few cases with effusions and may not be comparable to the cats with dry FIP. Assigning distinct control groups to each group with mixed or dry FIP would result in a small number of cases in each control group, making reliable statistical analysis difficult. No statistically significant difference was detected in age, body weight, and body temperature between the FIP and control groups, and we concluded that comparisons between the control and dry FIP groups were possible for parameters not related to pleural and abdominal effusions.

Another limitation is the lack of information regarding the therapeutic effects of constituent substances other than MT0901, which is known as the pharmacologically active ingredient of Mutian X [16]. MT0901 is believed to be the main active ingredient, but the efficacy and safety of other ingredients against FIP have not been elucidated to date. Moreover, although the active ingredient of the injectable Mutian product, Mutian II, has been already indicated to be identical to that of Mutian X for oral administration, according to the manufacturer’s information, it has not yet been demonstrated by any other third party [13]. However, this injectable nucleoside analogue has been already utilized similarly with oral administrative Mutian X in clinical cat therapy, resulting in little debate or doubt on their clinical use or outcomes [32,33]. Within the field of feline infectious disease, both formulations have been known widely as FIP treatment drugs with identical active ingredients; both formulations have already been used similarly by many cat owners around the world [14]. We have been in close contact with the manufacturers of Mutian X (oral administration) and Mutian II (injectable) to routinely ensure their correct dosage and method of administration for both formulations. We assume, however, that further data disclosure from the manufacturer is required in the future regarding the significance of ingredients other than MT0901 and the active ingredient of the injectable version. Furthermore, this is important as Mutian X and its main active ingredient will be officially approved in the field of veterinary medicine and supplied overtly to clinical sites worldwide.

Many FIP cats around the world are threatened by the disease, and veterinarians and owners are seeking information about Mutian. As such, if Mutian and the other related nucleic acid analogues with similar anti-coronavirus efficacy are evaluated in randomized controlled trials, companion diagnostics will help predict their efficacy. We believe that the advocacy of biomarker candidates, as suggested in our investigative study, is beneficial for the early evaluation of clinical trials, with resultant benefits for cats with FIP.

## 5. Conclusions

In this study, we demonstrated the apparent efficacy of Mutian treatment in client-owned cats with naturally occurring mixed and dry FIP. The therapeutic effects of Mutian administration on these two types of FIP were superior to those in the previously examined wet FIP cases. Thus, cats with effusive FIP (mixed as well as wet FIP) whose circulating TB levels were significantly elevated prior to initial drug administration may not respond satisfactorily to or even survive Mutian treatment, providing reasonable opportunities to undertake specific secondary measures, such as discontinuation of the drug or administration of alternative therapies at an earlier stage. Body temperature, appetite, and activity scores were significantly reduced prior to drug administration in the cats with mixed and dry FIP who further progressed to severe disease, showing the same tendency as those in the cats with wet FIP. In mixed and dry FIP, monitoring apparent neurological clinical signs at the initial examination can be expected to predict certain clinical benefits of Mutian use as a therapeutic agent, along with numerical indicators including body temperature, appetite, and activity scores.

## Figures and Tables

**Table 1 vetsci-10-00136-t001:** Statistical comparison of signalment and clinical parameters between cats with and without FIP.

Parameters	Cats without FIP			Cats with Mixed FIP			*p*-Value-1	Cats with Dry FIP			*p*-Value-2
	*n*	Mean	SE	*n*	Mean	SE		*n*	Mean	SE	
Age (months)	30	42.80	9.25	161	16.91	2.12	NS	163	17.41	1.68	NS
Body weight (kg)	29	3.05	0.25	161	2.62	0.08	NS	163	2.92	0.08	NS
Body temperature (℃)	24	38.51	0.20	142	38.69	0.12	NS	137	38.82	0.08	NS
Appetite score	30	4.17	0.29	161	3.00	0.11	<0.0005	163	3.53	0.13	NS
Activity score	30	4.20	0.29	161	3.03	0.09	<0.0003	163	3.72	0.12	NS
HCT (%)	28	32.94	1.73	161	23.21	0.55	<0.0001	163	29.62	0.67	NS
A/G	30	0.68	0.03	159	0.44	0.01	<0.0001	162	0.49	0.01	<0.0001
SAA (μg/mL)	28	72.51	17.23	158	111.38	5.76	<0.002	157	81.46	5.49	NS
TB (mg/dL)	14	0.76	0.20	122	1.64	0.18	NS	99	0.81	0.12	NS

Differences are judged to be statistically significant at *p* < 0.05, using Mann–Whitney non-parametric U-test. *p*-value-1 shows the statistical significance between the cats with mixed FIP and the control group, and *p*-value-2 shows the statistical significance between the cats with dry FIP and the control group. Abbreviations: FIP, feline infectious peritonitis; SE, standard error of the mean; NS, not significant; HCT, hematocrit; A/G, albumin-to-globulin ratio; SAA, serum amyloid-A; TB, total bilirubin.

**Table 2 vetsci-10-00136-t002:** Statistical comparison of categorical parameters between cats with and without FIP.

	Cats without FIP			Cats with Mixed FIP			*p*-Value-1	Cats with Dry FIP			*p*-Value-2
	*n*	Positive	Negative	*n*	Positive	Negative		*n*	Positive	Negative	
Diarrhea	30	12	18	161	36	125	NS	163	35	128	<0.04
Vomiting	30	7	23	161	16	145	NS	163	15	148	NS
Neurological signs	30	6	24	161	46	115	NS	163	43	120	NS
Ocular lesions	30	4	26	161	11	150	NS	163	30	133	NS
Abdominal lymphadenopathy	30	5	25	161	36	125	NS	163	63	100	<0.03
α1AG *	24	17	7	154	153	1	<0.0001	155	143	12	<0.006
PCR testing (blood)	26	3	23	156	138	18	<0.0001	153	124	29	<0.0001
Ascite retention	30	6	24	161	133	28	<0.0001	ND	ND	ND	
Pleural effusion retention	30	2	28	161	47	114	<0.02	ND	ND	ND	
PCR testing (ascites or pleural) **	4	0	4	113	108	5	<0.0001	ND	ND	ND	

Differences are judged to be statistically significant at *p* < 0.05, using Fisher’s exact test. *p*-value-1 shows the statistical significance between the cats with mixed FIP and the control group, and *p*-value-2 shows the statistical significance between the cats with dry FIP and the control group. * The samples were considered positive if the α1AG level exceeded the upper limit of the normal range (736 μg/mL). ** The samples were considered positive for FCoV when the target gene was detected in both or either ascites or pleural effusions. Abbreviations: FIP, feline infectious peritonitis; NS, not significant; PCR, polymerase chain reaction; α1AG, α1-acid glycoprotein; ND, not detected.

## Data Availability

The datasets used and/or analyzed during the current study are available from the corresponding author upon reasonable request.

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
