# Peer review of "Prognostic Prediction for Therapeutic Effects of Mutian on 324 Client-Owned Cats with Feline Infectious Peritonitis Based on Clinical Laboratory Indicators and Physical Signs"

_vetsci, 2023, doi:10.3390/vetsci10020136_

Round 1
Reviewer 1 Report
In this study, the authors report the effects of Mutian administration on clinical and laboratory parameters of cats with mixed or dry feline infectious peritonitis (FIP). The study presents some major drawbacks that cannot be easily addressed.
1. There is no evidence that Mutian X and Mutian II contains exactly the same compounds at the same concentrations and this could have biased the obtained results.
2. I could not find any authorization of the institutional ethics committee, which should advise about the ethical feasibility of the study.
3. There was no control group in the experiment. A group of cats not treated with Mutian, but with extant therapeutical protocol, should have been included for comparison.
4. There is no clear diagnosis of FIP for the treated cats. FIP diagnosis is complicated and the authors did not provide any information on how they reached this diagnosis.
Author Response
Response to the Reviewer #1
In this study, the authors report the effects of Mutian administration on clinical and laboratory parameters of cats with mixed or dry feline infectious peritonitis (FIP). The study presents some major drawbacks that cannot be easily addressed.
- There is no evidence that Mutian X and Mutian II contains exactly the same compounds at the same concentrations and this could have biased the obtained results.
Responses:
First of all, combined use of Mutian II (injectable) and Mutian X (orally-administrative) to several cats has been already reported by the famous expert in FIP therapy without imposing doubts or debates on their clinical use or outcomes at that moment (Addie DD, et al. 2020). Later than this study, it is an almost obvious that both formulations have been known widely as FIP treatment drugs with the identical active ingredients, considering the fact that both of them has been already used similarly by many cat owners around the world (Jones S, et al. 2021).
In addition, we promise that we has been in close contact with manufacturers of Mutian X (orally-administrative) and Mutian II (injectable) to routinely ensure their correct dosage and method of administration for both formulations. These therapeutic processes has been performed not only in the treatments described in our current report but also already in those published in Veterinary Sciences (2021, 8, 328). Although administration routes for these two kinds of formulations are different because they are tablets and liquid forms, it is quite natural for us to strictly adhere to the same daily dosage of the main active ingredients per body weight. Including the dose control, we have continued to provide safe treatments with Mutian by providing advance explanations of pros and cons to the cat’s owners about both formulations.
Finally, I have already received an official document from the manufacturer that certifies that the active ingredients of both formulations are identical, and recognize that there is no concern about the issue pointed out by the reviewer #1.
As the reviewer claimed, it is true that the any accurate scientific analysis have not been reported yet on this point by a third party. Therefore, I have explained the above points as limitation of our study in the revised manuscript (line 486-493, in our revised manuscript).
- I could not find any authorization of the institutional ethics committee, which should advise about the ethical feasibility of the study.
Responses:
All of the cats presented in our current report were completely excluded from receiving placebo, which may occur potentially in a randomized controlled trial (RCT). Additionally, I have assured in advance to their owners that real Mutian would be administered to all of the FIP cats under optimal conditions for their treatment. Therefore, I can state that administrations of Mutian to the cats with FIP were recognized by the owners as standard of care and not as a RCT or experimental therapy.
Ethical review and approval were waived because the above informed consent acquisition was legally implemented in accordance with the declaration of the Japanese Veterinary Medical Association and all of the data were obtained within the scope of usual veterinary care and anonymized appropriately.
These comments were briefly described in the Institutional Review Board Statement, but were not included in the text of the original manuscript. Although it becomes in duplicate, their summary has been described in the text part of the revised manuscript for the reader’s understanding (line 137-143, in our revised manuscript).
- There was no control group in the experiment. A group of cats not treated with Mutian, but with extant therapeutical protocol, should have been included for comparison.
Responses:
As I described in the above section, this study is not a RCT and not to compare the efficacy of Mutian for FIP with those of untreated or other extant treatments. Thus, no control group of FIP cats was set in our present study.
Considering the disease severity and high mortality rate in the short term after disease onset reported in several previous studies, also with Mutian’s apparent therapeutic effect on FIP, I expect it may be difficult to recruit the disease cats into a control group. I suppose some kind of ingenuity will be required in this regard, if a RCT is to be conducted in future.
- There is no clear diagnosis of FIP for the treated cats. FIP diagnosis is complicated and the authors did not provide any information on how they reached this diagnosis.
Responses:
Diseases for all of the cats including those with FIP, were diagnosed basically according to the method described in our previous report (Vet. Sci., 2021, 8, 328), and the similar explanations have been already described in our present report.
As the reviewer has mentioned, obtaining a definitive diagnosis of FIP based on non-invasive approaches is difficult. Under such a circumstance, in order to make a diagnosis of FIP in our routine healthcare, we made a diagnosis by comprehensively considering the above indicators. I have added some descriptions in our revised manuscript for the reader's further understanding (line 184-188, in our revised manuscript).
Reviewer 2 Report
The main question addressed by the research is to determine whether the therapeutic effect of Mutian on dry and mixed FIP can be predicted using clinical indicators before starting treatment.
The topic is original or relevant in the field; Tt addresses a specific gap in the field.
It adds to the subject area compared with other published material the use of Mutian on dry and mixed FIP.
No other specific improvements should be considered.
No further controls should be considered.
The conclusions consistent with the evidence and arguments presented and do they address the main question posed.
References are appropriate.
No additional comments on the tables and figures are needed.
Author Response
Response to the Reviewer #2
I would like to thank you for kind comments. I understand that you have judged our original manuscript to be acceptable for publication. Thanks again for your cooperation.
Masato Katayama/ Corresponding author
Reviewer 3 Report
Comments on the Abstract:
Lines 9 and 10: Sentence could be more fluent.
Line 12: a comma instead of a period after the word "treatment" would give a clearer meaning to the two sentences.
Line 16: reconsider replacing "who" with "that" and wrong use of comma after "84 days".
Line 22: In a list of three or more items, use a comma before the conjunction separating the final item. Articles, like a, an, and the, show noun specificity and shouldn't be forgotten when writing. ("as the presence").
Line 40: Subjects and verbs should match in person (first, second, or third) and number (singular or plural): reconsider replacing "are" with "is" .
Lines 40-41: Sentence could be more fluent.
Lines 65-67: Sentence could be more fluent ( "by third parties" instead of "by the third parties"; "is assumingly" instead of "assumingly").
Line 90: comma after "slow movements".
Lines 91 and 92: Two independent clauses joined by a coordinating conjunction must include a comma before the conjunction: comma after "simultaneously". It can be considered "and their staging follows dry-type criteria" instead of "and staging of them follows dry-type criteria".
Line 96: Article "an" shows noun specificity and shouldn't be forgotten: "an inability" instead of "inability".
Line 138: "through a comprehensive examination" instead of "through comprehensive examination"
Line 194: "in the dry FIP cats" instead of "in the dry FIP"
Line 245: "Figures 1 and 2" instead of "Figure 1 and 2"
Lines 246 and 247: "only one patient" instead of "only one patients"
Line 248: "only for" instead of "only during"
Line 255: Make it fluent (e.g. Statistical comparison of dry FIP signaling and clinical parameters in surviving and non-surviving cats).
Line 260 : see the previous suggestion (line 255)
Lines 256-269: sentence too long and hard to understand
Line 331, Lines 333 and 334, Line 421: "Figures 1 and 2" instead of "Figure 1 and 2"
Line 410: Make it fluent. (e.g., "The majority of neurological FIP is caused by FCoV-induced hydrocephalus")
Lines 419-421: the sentence is difficult to understand ("the reductions in SAA levels" instead of "the reduction in SAA levels"?)
Lines 434-435: Make it fluent (e.g., "The current control group contained a few cases with effusions and may not be comparable to the cats with dry FIP.")
Line 438: comma after "body weight"
Lines 447-448: Make it fluent (e.g., "In clinical cat therapy, this injectable nucleoside analogue has already been used in conjunction with oral administration Mutian X [32].")
Line 450: "injectable" instead of "injective"
Line 451: Remove the comma after "active ingredient"
Lines 455-459: Additional commas and some writing errors make it difficult to read the sentences (e.g., "As such, if Mutian and the other related nucleic acid analogues with similar anti-coronavirus efficacy are evaluated in randomised controlled trials, companion diagnostics will help predict their efficacy. We believe that the advocacy of biomarker candidates, as suggested in our investigative study, is beneficial for the early evaluation of clinical trials, with resultant benefits for cats with FIP.")
Author Response
Reviewer #3: Comments and Suggestions for Authors
Comments on the Abstract:
- Lines 9 and 10: Sentence could be more fluent.
Response:
This sentence has been amended without repeated use of ‘type’ (line 32-34, in the revised manuscript).
- Line 12: a comma instead of a period after the word "treatment" would give a clearer meaning to the two sentences.
Response:
A comma was used (line 35, in the revised manuscript).
- Line 16: reconsider replacing "who" with "that" and wrong use of comma after "84 days".
Response:
I have replaced “who” with “that” and removed a comma (line 39, in the revised manuscript).
- Line 22: In a list of three or more items, use a comma before the conjunction separating the final item. Articles, like a, an, and the, show noun specificity and shouldn't be forgotten when writing. ("as the presence").
Response:
I have entered a comma and the article of ‘the’ appropriately according to the reviewer’s suggestion (line 45-46, in the revised manuscript).
- Line 40: Subjects and verbs should match in person (first, second, or third) and number (singular or plural): reconsider replacing "are" with "is" .
Response:
I have replaced “are” with “is” according to the reviewer’s suggestion (line 64, in the revised manuscript).
- Lines 40-41: Sentence could be more fluent.
Response:
I have simplified the sentences to be more fluent (line 63-65, in the revised manuscript).
- Lines 65-67: Sentence could be more fluent ( "by third parties" instead of "by the third parties"; "is assumingly" instead of "assumingly").
Response:
I have simplified the sentences to be more fluent (line 90-91, in the revised manuscript).
- Line 90: comma after "slow movements".
Response:
A comma was added in our revised manuscript (line 114, in the revised manuscript).
- Lines 91 and 92: Two independent clauses joined by a coordinating conjunction must include a comma before the conjunction: comma after "simultaneously". It can be considered "and their staging follows dry-type criteria" instead of "and staging of them follows dry-type criteria".
Response:
I have added a comma after “simultaneously” and change the description according to the reviewer’s suggestion (line 115, in the revised manuscript).
- Line 96: Article "an" shows noun specificity and shouldn't be forgotten: "an inability" instead of "inability".
Response:
I have added the article of “an” before “inability” (line 120, in the revised manuscript).
- Line 138: "through a comprehensive examination" instead of "through comprehensive examination"
Response:
I have revised the description according to the reviewer’s suggestion (line 170, in the revised manuscript).
- Line 194: "in the dry FIP cats" instead of "in the dry FIP"
Response:
I have revised the description according to the reviewer’s suggestion (line 230, in the revised manuscript).
- Line 245: "Figures 1 and 2" instead of "Figure 1 and 2"
Response:
I have revised the description according to the reviewer’s suggestion (line 283, in the revised manuscript).
- Lines 246 and 247: "only one patient" instead of "only one patients"
Response:
I have revised the description according to the reviewer’s suggestion (line 285, in the revised manuscript).
- Line 248: "only for" instead of "only during"
Response:
I have revised the description according to the reviewer’s suggestion (line 287, in the revised manuscript).
- Line 255: Make it fluent (e.g. Statistical comparison of dry FIP signaling and clinical parameters in surviving and non-surviving cats).
Response:
I was confused by the reviewer’s suggestion that only the title of Table 5 should be revised, but finally decided that it is necessary to be consistent with the titles of the other Tables (Table 3 and 4), so I have left it.
- Line 260 : see the previous suggestion (line 255)
Response:
I decided that it is necessary to be consistent with the titles of the other Tables (Table 3 and 4), so I have left it.
- Lines 256-269: sentence too long and hard to understand
Response:
The descriptions between lines 256-258 and those between lines 261-263 are the Footnote for Tables 5 and 6, respectively. I believe all of the readers possible to understand the descriptions between lines 265-269.
- Line 331, Lines 333 and 334, Line 421: "Figures 1 and 2" instead of "Figure 1 and 2"
Response:
I have revised the description according to the reviewer’s suggestion (line 369, 371, 372, and 459, in the revised manuscript).
- Line 410: Make it fluent. (e.g., "The majority of neurological FIP is caused by FCoV-induced hydrocephalus")
Response:
I have revised the description according to the reviewer’s suggestion (line 448-449, in the revised manuscript).
- Lines 419-421: the sentence is difficult to understand ("the reductions in SAA levels" instead of "the reduction in SAA levels"?)
Response:
I have revised the description according to the reviewer’s suggestion (line 458, in the revised manuscript).
- Lines 434-435: Make it fluent (e.g., "The current control group contained a few cases with effusions and may not be comparable to the cats with dry FIP.")
Response:
I have revised the description according to the reviewer’s suggestion (line 472-473, in the revised manuscript).
- Line 438: comma after "body weight"
Response:
I have added a comma according to the reviewer’s suggestion (line 476, in the revised manuscript).
- Lines 447-448: Make it fluent (e.g., "In clinical cat therapy, this injectable nucleoside analogue has already been used in conjunction with oral administration Mutian X [32].")
Response:
We wished to explain that Mutian II (injectable) is used as compatible with Mutian X (oral-administrative), not in conjunction. I've already added a lot of clarifications in response to the Reviewer#1's inquiry into this section, so I hope you can understand my intension (line 486-493, in the revised manuscript).
- Line 450: "injectable" instead of "injective"
Response:
I have revised the description according to the reviewer’s suggestion (line 495, in the revised manuscript).
- Line 451: Remove the comma after "active ingredient"
Response:
I have removed the comma according to the reviewer’s suggestion (line 496, in the revised manuscript).
- Lines 455-459: Additional commas and some writing errors make it difficult to read the sentences (e.g., "As such, if Mutian and the other related nucleic acid analogues with similar anti-coronavirus efficacy are evaluated in randomised controlled trials, companion diagnostics will help predict their efficacy. We believe that the advocacy of biomarker candidates, as suggested in our investigative study, is beneficial for the early evaluation of clinical trials, with resultant benefits for cats with FIP.")
Response:
I have revised the description according to the reviewer’s suggestion (line 500-503, in the revised manuscript).
Reviewer 4 Report
This manuscript is a well written description of a study examining the benefit of various indicators to determine the therapeutic effects of a treatment for FIP. I have only a couple of concerns. The control group consisted of 30 cats suspected of having FIP but eventually diagnosed with a disease other than FIP. There is a lack of detail regarding the initial diagnosis (or suspicion of FIP), how FIP was eventually ruled out and whether the cats were otherwise treated exactly the same as the other cats in the study. Did they receive any Mutian treatment? The PCR method is referenced but some details should be provided such as the target of the primers. Also, the PCR should be defined as a reverse transcriptase PCR to distinguish it from real-time PCRs.
minor comment:
Line 266 should be "activity score and clinical laboratory "
Line 313 if the relapsing cats that were treated were further evaluated the results should be included.
Author Response
Reviewer #4: Comments and Suggestions for Authors
This manuscript is a well written description of a study examining the benefit of various indicators to determine the therapeutic effects of a treatment for FIP. I have only a couple of concerns. The control group consisted of 30 cats suspected of having FIP but eventually diagnosed with a disease other than FIP. There is a lack of detail regarding the initial diagnosis (or suspicion of FIP), how FIP was eventually ruled out and whether the cats were otherwise treated exactly the same as the other cats in the study. Did they receive any Mutian treatment? The PCR method is referenced but some details should be provided such as the target of the primers. Also, the PCR should be defined as a reverse transcriptase PCR to distinguish it from real-time PCRs.
Response:
In response to the Reviewer #1's inquiry No.4, I added a new description of diagnosis for FIP in the revised manuscript (line 184-187, in the revised manuscript). Because the 30 cats with non-FIP diseases were judged not to have FIP during the diagnostic process described in this section, this description was also added to the revised manuscript (line 187-188, in the revised manuscript).
A few of the non-FIP cats received Mutian treatments just for a short period, but all data of the non-FIP cats were obtained prior to their initial therapy and it is reasonably understood that those data are independent of any treatments including administration of Mutian.
The viral gene to be analyzed by a reverse transcriptase PCR (RT-PCR) technology and its method are described in Reference No.18 in the original manuscript (Tanaka Y, et al., BMC Vet. Res., 2015), already cited in our previous report (Vet. Sci., 2021). We have outsourced the analysis to an external contract research organization (Canine Lab Co., Ltd., Tokyo, Japan), and as the technology is highly confidential, we were unfortunately not able to obtain any information other than the cited papers (Reference No.18). The detail of the abbreviations for RT-PCR has been described at its first appearance in the abstract and the main text of our revised manuscript (line 36 and 150, in the revised manuscript).
Minor comment:
- Line 266 should be "activity score and clinical laboratory"
Response:
I have revised the description according to the reviewer’s suggestion (line 304, in the revised manuscript).
- Line 313 if the relapsing cats that were treated were further evaluated the results should be included.
Response:
In our present report, we considered that most of the veterinarians have an interest in the frequency of recurrence in surviving cats after Mutian treatment, and have just disclosed it as reference data. We have decided, however, follow-up of these cases are outside the scope of this study, and are planning to evaluate them independently in another case report in the future.